# Bayesian Mixture Modeling and Inference based Thompson Sampling in Monte-Carlo Tree Search

**Aijun Bai**
Univ. of Sci. & Tech. of China
baj@mail.ustc.edu.cn

**Feng Wu**
University of Southampton
fw6e11@ecs.soton.ac.uk

**Xiaoping Chen**
Univ. of Sci. & Tech. of China
xpchen@ustc.edu.cn

## Abstract

Monte-Carlo tree search (MCTS) has been drawing great interest in recent years for planning and learning under uncertainty. One of the key challenges is the trade-off between exploration and exploitation. To address this, we present a novel approach for MCTS using Bayesian mixture modeling and inference based Thompson sampling and apply it to the problem of online planning in MDPs. Our algorithm, named Dirichlet-NormalGamma MCTS (DNG-MCTS), models the uncertainty of the accumulated reward for actions in the search tree as a mixture of Normal distributions. We perform inferences on the mixture in Bayesian settings by choosing conjugate priors in the form of combinations of Dirichlet and NormalGamma distributions and select the best action at each decision node using Thompson sampling. Experimental results confirm that our algorithm advances the state-of-the-art UCT approach with better values on several benchmark problems.

## 1 Introduction

*Markov decision processes* (MDPs) provide a general framework for planning and learning under uncertainty. We consider the problem of online planning in MDPs without prior knowledge on the underlying transition probabilities. *Monte-Carlo tree search* (MCTS) can find near-optimal policies in our domains by combining tree search methods with sampling techniques. The key idea is to iteratively evaluate each state in a best-first search tree by the mean outcome of simulation samples. It is model-free and requires only a black-box simulator (generative model) of the underlying problems. To date, great success has been achieved by MCTS in variety of domains, such as game play [1, 2], planning under uncertainty [3, 4, 5], and Bayesian reinforcement learning [6, 7].

When applying MCTS, one of the fundamental challenges is the so-called *exploration versus exploitation* dilemma: an agent must not only exploit by selecting the best action based on the current information, but should also keep exploring other actions for possible higher future payoffs. Thompson sampling is one of the earliest heuristics to address this dilemma in *multi-armed bandit problems* (MABs) according to the principle of *randomized probability matching* [8]. The basic idea is to select actions stochastically, based on the probabilities of being optimal. It has recently been shown to perform very well in MABs both empirically [9] and theoretically [10]. It has been proved that Thompson sampling algorithm achieves logarithmic expected regret which is asymptotically optimal for MABs. Comparing to the UCB1 heuristic [3], the main advantage of Thompson sampling is that it allows more robust convergence under a wide range of problem settings.

In this paper, we borrow the idea of Thompson sampling and propose the Dirichlet-NormalGamma MCTS (DNG-MCTS) algorithm — a novel Bayesian mixture modeling and inference based Thompson sampling approach for online planning in MDPs. In this algorithm, we use a mixture of Normal distributions to model the unknown distribution of the accumulated reward of performing a particular action in the MCTS search tree. In the present of online planning for MDPs, a conjugate prior

exists in the form of a combination of Dirichlet and NormalGamma distributions. By choosing the conjugate prior, it is then relatively simple to compute the posterior distribution after each accumulated reward is observed by simulation in the search tree. Thompson sampling is then used to select the action to be performed by simulation at each decision node. We have tested our DNG-MCTS algorithm and compared it with the popular UCT algorithm in several benchmark problems. Experimental results show that our proposed algorithm has outperformed the state-of-the-art for online planning in general MDPs. Furthermore, we show the convergence of our algorithm, confirming its technical soundness.

The reminder of this paper is organized as follows. In Section 2, we briefly introduce the necessary background. Section 3 presents our main results — the DNG-MCTS algorithm. We show experimental results on several benchmark problems in Section 4. Finally in Section 5 the paper is concluded with a summary of our contributions and future work.

## 2  Background

In this section, we briefly review the MDP model, the MAB problem, the MCTS framework, and the UCT algorithm as the basis of our algorithm. Some related work is also presented.

### 2.1  MDPs and MABs

Formally, an MDP is defined as a tuple $\langle S, A, T, R \rangle$, where $S$ is the state space, $A$ is the action space, $T(s'|s, a)$ is the probability of reaching state $s'$ if action $a$ is applied in state $s$, and $R(s, a)$ is the reward received by the agent. A *policy* is a decision rule mapping from states to actions and specifying which action should be taken in each state. The aim of solving an MDP is to find the *optimal* policy $\pi$ that maximizes the expected reward defined as $V_\pi(s) = \mathbb{E}[\sum_{t=0}^{H} \gamma^t R(s_t, \pi(s_t))]$, where $H$ is the planing horizon, $\gamma \in (0, 1]$ is the discount factor, $s_t$ is the state in time step $t$ and $\pi(s_t)$ is the action selected by policy $\pi$ in state $s_t$.

Intuitively, an MAB can be seen as an MDP with only one state $s$ and a stochastic reward function $R(s, a) := X_a$, where $X_a$ is a random variable following an unknown distribution $f_{X_a}(x)$. At each time step $t$, one action $a_t$ must be chosen and executed. A stochastic reward $X_{a_t}$ is then received accordingly. The goal is to find a sequence of actions that minimizes the cumulative regret defined as $R_T = \mathbb{E}[\sum_{t=1}^{T} (X_{a^*} - X_{a_t})]$, where $a^*$ is the true best action.

### 2.2  MCTS and UCT

To solve MDPs, MCTS iteratively evaluates a state by: (1) selecting an action based on a given *action selection* strategy; (2) performing the selected action by Monte-Carlo simulation; (3) recursively evaluating the resulted state if it is already in the search tree, or inserting it into the search tree and running a *rollout policy* by simulations. This process is applied to descend through the search tree until some terminate conditions are reached. The simulation result is then back-propagated through the selected nodes to update their statistics.

The UCT algorithm is a popular approach based on MCTS for planning under uncertainty [3]. It treats each state of the search tree as an MAB, and selects the action that maximizes the UCB1 heuristic $\bar{Q}(s, a) + c\sqrt{\log N(s)/N(s, a)}$, where $\bar{Q}(s, a)$ is the mean return of action $a$ in state $s$ from all previous simulations, $N(s, a)$ is the visitation count of action $a$ in state $s$, $N(s)$ is the overall count $N(s) = \sum_{a \in A} N(s, a)$, and $c$ is the exploration constant that determines the relative ratio of exploration to exploitation. It is proved that with an appropriate choice of $c$ the probability of selecting the optimal action converges to 1 as the number of samples grows to infinity.

### 2.3  Related Work

The fundamental assumption of our algorithm is modeling unknown distribution of the accumulated reward for each state-action pair in the search tree as a mixture of Normal distributions. A similar assumption has been made in [11], where they assumed a Normal distribution over the rewards. Comparing to their approach, as we will show in Section 3, our assumption on Normal mixture is more realistic for our problems. Tesauro *et al.*[12] developed a Bayesian UCT approach to MCTS

using Gaussian approximation. Specifically, their method propagates probability distributions of rewards from leaf nodes up to the root node by applying MAX (or MIN) extremum distribution operator for the interior nodes. Then, it uses modified UCB1 heuristics to select actions on the basis of the interior distributions. However, extremum distribution operation on decision nodes is very time-consuming because it must consider over all the child nodes. In contrast, we treat each decision node in the search tree as an MAB, maintain a posterior distribution over the accumulated reward for each applicable actions separately, and then select the best action using Thompson sampling.

## 3  The DNG-MCTS Algorithm

This section presents our main results — a *Bayesian mixture modeling and inference based Thompson sampling* approach for MCTS (DNG-MCTS).

### 3.1  The Assumptions

For a given MDP policy $\pi$, let $X_{s,\pi}$ be a random variable that denotes the accumulated reward of following policy $\pi$ starting from state $s$, and $X_{s,a,\pi}$ denotes the accumulated reward of first performing action $a$ in state $s$ and then following policy $\pi$ thereafter. Our assumptions are: (1) $X_{s,\pi}$ is sampled from a Normal distribution, and (2) $X_{s,a,\pi}$ can be modeled as a mixture of Normal distributions. These are realistic approximations for our problems with the following reasons.

Given policy $\pi$, an MDP reduces to a Markov chain $\{s_t\}$ with finite state space $S$ and the transition function $T(s'|s, \pi(s))$. Suppose that the resulting chain $\{s_t\}$ is ergodic. That is, it is possible to go from every state to every other state (not necessarily in one move). Let $w$ denote the stationary distribution of $\{s_t\}$. According to the central limit theorem on Markov chains [13, 14], for any bounded function $f$ on the state space $S$, we have:

$$\frac{1}{\sqrt{n}}(\sum_{t=0}^{n} f(s_t) - n\mu) \to N(0, \sigma^2) \text{ as } n \to \infty, \tag{1}$$

where $\mu = \mathbb{E}_w[f]$ and $\sigma$ is a constant depending only on $f$ and $w$. This indicates that the sum of $f(s_t)$ follows $N(n\mu, n\sigma^2)$ as $n$ grows to infinity. It is then natural to approximate the distribution of $\sum_{t=0}^{n} f(s_t)$ as a Normal distribution if $n$ is sufficiently large.

Considering finite-horizon MDPs with horizon $H$, if $\gamma = 1$, $X_{s_0,\pi} = \sum_{t=0}^{H} R(s_t, \pi(s_t))$ is a sum of $f(s_t) = R(s_t, \pi(s_t))$. Thus, $X_{s_0,\pi}$ is approximately normally distributed for each $s_0 \in S$ if $H$ is sufficiently large. On the other hand, if $\gamma \neq 1$, $X_{s_0,\pi} = \sum_{t=0}^{H} \gamma^t R(s_t, \pi(s_t))$ can be rewritten as a linear combination of $\sum_{t=0}^{n} f(s_t)$ for $n = 0$ to $H$ as follow:

$$X_{s_0,\pi} = (1 - \gamma) \sum_{n=0}^{H-1} \gamma^n \sum_{t=0}^{n} f(s_t) + \gamma^H \sum_{t=0}^{H} f(s_t) \tag{2}$$

Notice that a linear combination of independent or correlated normally distributed random variables is still normally distributed. If $H$ is sufficiently large and $\gamma$ is close to 1, it is reasonable to approximate $X_{s_0,\pi}$ as a Normal distribution. Therefore, we assume that $X_{s,\pi}$ is normally distributed in both cases.

If the policy $\pi$ is not fixed and may change over time (e.g., the derived policy of an online algorithm before it converges), the real distribution of $X_{s,\pi}$ is actually unknown and could be very complex. However, if the algorithm is guaranteed to converge in the limit (as explained in Section 3.5, this holds for our DNG-MCTS algorithm), it is convenient and reasonable to approximate $X_{s,\pi}$ as a Normal distribution.

Now consider the accumulated reward of first performing action $a$ in $s$ and following policy $\pi$ thereafter. By definition, $X_{s,a,\pi} = R(s, a) + \gamma X_{s',\pi}$, where $s'$ is the next state distributed according to $T(s'|s, a)$. Let $Y_{s,a,\pi}$ be a random variable defined as $Y_{s,a,\pi} = (X_{s,a,\pi} - R(s, a))/\gamma$. We can see that the pdf of $Y_{s,a,\pi}$ is a convex combination of the pdfs of $X_{s',\pi}$ for each $s' \in S$. Specifically, we have $f_{Y_{s,a,\pi}}(y) = \sum_{s' \in S} T(s'|s, a) f_{X_{s',\pi}}(y)$. Hence it is straightforward to model the distribution of $Y_{s,a,\pi}$ as a mixture of Normal distributions if $X_{s',\pi}$ is assumed to be normally distributed for each $s' \in S$. Since $X_{s,a,\pi}$ is a linear function of $Y_{s,a,\pi}$, $X_{s,a,\pi}$ is also a mixture of Normal distributions under our assumptions.

## 3.2 The Modeling and Inference Methods

In Bayesian settings, the unknown distribution of a random variable $X$ can be modeled as a parametric likelihood function $L(x|\theta)$ depending on the parameters $\theta$. Given a prior distribution $P(\theta)$, and a set of past observations $Z = \{x_1, x_2, \dots \}$, the posterior distribution of $\theta$ can then be obtained using Bayes' rules: $P(\theta|Z) \propto \prod_i L(x_i|\theta)P(\theta)$.

Assumption (1) implies that it suffices to model the distribution of $X_{s,\pi}$ as a Normal likelihood $N(\mu_s, 1/\tau_s)$ with unknown mean $\mu_s$ and precision $\tau_s$. The precision is defined as the reciprocal of the variance, $\tau = 1/\sigma^2$. This is chosen for mathematical convenience of introducing the NomralGamma distribution as a conjugate prior. A NormalGamma distribution is defined by the hyper-parameters $\langle \mu_0, \lambda, \alpha, \beta \rangle$ with $\lambda > 0$, $\alpha \geq 1$ and $\beta \geq 0$. It is said that $(\mu, \tau)$ follows a NormalGamma distribution $NormalGamma(\mu_0, \lambda, \alpha, \beta)$ if the pdf of $(\mu, \tau)$ has the form

$$f(\mu, \tau | \mu_0, \lambda, \alpha, \beta) = \frac{\beta^\alpha \sqrt{\lambda}}{\Gamma(\alpha)\sqrt{2\pi}} \tau^{\alpha - \frac{1}{2}} e^{-\beta\tau} e^{-\frac{\lambda\tau(\mu - \mu_0)^2}{2}}. \tag{3}$$

By definition, the marginal distribution over $\tau$ is a Gamma distribution, $\tau \sim Gamma(\alpha, \beta)$, and the conditional distribution over $\mu$ given $\tau$ is a Normal distribution, $\mu \sim N(\mu_0, 1/(\lambda\tau))$.

Let us briefly recall the posterior of $(\mu, \tau)$. Suppose $X$ is normally distributed with unknown mean $\mu$ and precision $\tau$, $x \sim N(\mu, 1/\tau)$, and that the prior distribution of $(\mu, \tau)$ has a NormalGamma distribution, $(\mu, \tau) \sim NormalGamma(\mu_0, \lambda_0, \alpha_0, \beta_0)$. After observing $n$ independent samples of $X$, denoted $\{x_1, x_2, \dots, x_n\}$, according to the Bayes' theorem, the posterior distribution of $(\mu, \tau)$ is also a NormalGamma distribution, $(\mu, \tau) \sim NormalGamma(\mu_n, \lambda_n, \alpha_n, \beta_n)$, where $\mu_n = (\lambda_0\mu_0 + n\bar{x})/(\lambda_0 + n), \lambda_n = \lambda_0 + n, \alpha_n = \alpha_0 + n/2$ and $\beta_n = \beta_0 + (ns + \lambda_0 n(\bar{x} - \mu_0)^2/(\lambda_0 + n))/2$, where $\bar{x} = \sum_{i=1}^{n} x_i/n$ is the sample mean and $s = \sum_{i=1}^{n}(x_i - \bar{x})^2/n$ is the sample variance.

Based on Assumption 2, the distribution of $Y_{s,a,\pi}$ can be modeled as a mixture of Normal distributions $Y_{s,a,\pi} = (X_{s,a,\pi} - R(s,a))/\gamma \sim \sum_{s' \in S} w_{s,a,s'} N(\mu_{s'}, 1/\tau_{s'})$, where $w_{s,a,s'} = T(s'|s,a)$ are the mixture weights such that $w_{s,a,s'} \geq 0$ and $\sum_{s' \in S} w_{s,a,s'} = 1$, which are previously unknown in Monte-Carlo settings. A natural representation on these unknown weights is via Dirichlet distributions, since Dirichlet distribution is the conjugate prior of a general discrete probability distribution. For state $s$ and action $a$, a Dirichlet distribution, denoted $Dir(\boldsymbol{\rho}_{s,a})$ where $\boldsymbol{\rho}_{s,a} = (\rho_{s,a,s_1}, \rho_{s,a,s_2}, \cdots)$, gives the posterior distribution of $T(s'|s,a)$ for each $s' \in S$ if the transition to $s'$ has been observed $\rho_{s,a,s'} - 1$ times. After observing a transition $(s,a) \rightarrow s'$, the posterior distribution is also Dirichlet and can simply be updated as $\rho_{s,a,s'} \leftarrow \rho_{s,a,s'} + 1$.

Therefore, to model the distribution of $X_{s,\pi}$ and $X_{s,a,\pi}$ we only need to maintain a set of hyper-parameters $\langle \mu_{s,0}, \lambda_s, \alpha_s, \beta_s \rangle$ and $\boldsymbol{\rho}_{s,a}$ for each state $s$ and action $a$ encountered in the MCTS search tree and update them by using Bayes' rules.

Now we turn to the question of how to choose the priors by initializing hyper-parameters. While the impact of the prior tends to be negligible in the limit, its choice is important especially when only a small amount of data has been observed. In general, priors should reflect available knowledge of the hidden model.

In the absence of any knowledge, *uninformative priors* may be preferred. According to the principle of indifference, uninformative priors assign equal probabilities to all possibilities. For NormalGamma priors, we hope that the sampled distribution of $\mu$ given $\tau$, i.e., $N(\mu_0, 1/(\lambda\tau))$, is as flat as possible. This implies an infinite variance $1/(\lambda\tau) \rightarrow \infty$, so that $\lambda\tau \rightarrow 0$. Recall that $\tau$ follows a Gamma distribution $Gamma(\alpha, \beta)$ with expectation $E[\tau] = \alpha/\beta$, so we have in expectation $\lambda\alpha/\beta \rightarrow 0$. Considering the parameter space ($\lambda > 0, \alpha \geq 1, \beta \geq 0$), we can choose $\lambda$ small enough, $\alpha = 1$ and $\beta$ sufficiently large to approximate this condition. Second, we hope the sampled distribution is in the middle of axis, so $\mu_0 = 0$ seems to be a good selection. It is worth noting that intuitively $\beta$ should not be set too large, or the convergence process may be very slow. For Dirichlet priors, it is common to set $\rho_{s,a,s'} = \delta$ where $\delta$ is a small enough positive for each $s \in S, a \in A$ and $s' \in S$ encountered in the search tree to have uninformative priors.

On the other hand, if some prior knowledge is available, *informative priors* may be preferred. By exploiting domain knowledge, a state node can be initialized with informative priors indicating its priority over other states. In DNG-MCTS, this is done by setting the hyper-parameters based

on subjective estimation for states. According to the interpretation of hyper-parameters of NormalGamma distribution in terms of pseudo-observations, if one has a prior mean of $\mu_0$ from $\lambda$ samples and a prior precision of $\alpha/\beta$ from $2\alpha$ samples, the prior distribution over $\mu$ and $\tau$ is $NormalGamma(\mu_0, \lambda, \alpha, \beta)$, providing a straightforward way to initialize the hyper-parameters if some prior knowledge (such as historical data of past observations) is available. Specifying detailed priors based on prior knowledge for particular domains is beyond the scope of this paper. The ability to include prior information provides important flexibility and can be considered an advantage of the approach.

### 3.3   The Action Selection Strategy

In DNG-MCTS, action selection strategy is derived using Thompson sampling. Specifically, in general Bayesian settings, action $a$ is chosen with probability:

$$P(a) = \int \mathbf{1} \left[ a = \underset{a'}{\arg\max}\, \mathbb{E}\left[ X_{a'} | \theta_{a'} \right] \right] \prod_{a'} P_{a'}(\theta_{a'} | Z)\, \mathrm{d}\boldsymbol{\theta} \qquad (4)$$

where $\mathbf{1}$ is the indicator function, $\theta_a$ is the hidden parameter prescribing the underlying distribution of reward by applying $a$, $\mathbb{E}[X_a|\theta_a] = \int x L_a(x|\theta_a)\,\mathrm{d}x$ is the expectation of $X_a$ given $\theta_a$, and $\boldsymbol{\theta} = (\theta_{a_1}, \theta_{a_2}, \dots)$ is the vector of parameters for all actions. Fortunately, this can efficiently be approached by sampling method. To this end, a set of parameters $\theta_a$ is sampled according to the posterior distributions $P_a(\theta_a|Z)$ for each $a \in A$, and the action $a^* = \arg\max_a \mathbb{E}[X_a|\theta_a]$ with highest expectation is selected.

In our implementation, at each decision node $s$ of the search tree, we sample the mean $\mu_{s'}$ and mixture weights $w_{s,a,s'}$ according to $NormalGamma(\mu_{s',0}, \lambda_{s'}, \alpha_{s'}, \beta_{s'})$ and $Dir(\boldsymbol{\rho}_{s,a})$ respectively for each possible next state $s' \in S$. The expectation of $X_{s,a,\pi}$ is then computed as $R(s,a) + \gamma \sum_{s' \in S} w_{s,a,s'} \mu_{s'}$. The action with highest expectation is then selected to be performed in simulation.

### 3.4   The Main Algorithm

The main process of DNG-MCTS is outlined in Figure 1. It is worth noting that the function ThompsonSampling has a boolean parameter $sampling$. If $sampling$ is true, Thompson sampling method is used to select the best action as explained in Section 3.3, otherwise a greedy action is returned with respect to the current expected transition probabilities and accumulated rewards of next states, which are $\mathbb{E}[w_{s,a,s'}] = \rho_{s,a,s'} / \sum_{x \in S} \rho_{s,a,x}$ and $\mathbb{E}[X_{s,\pi}] = \mu_{s,0}$ respectively.

At each iteration, the function DNG-MCTS uses Thompson sampling to recursively select actions to be executed by simulation from the root node to leaf nodes through the existing search tree $T$. It inserts each newly visited node into the tree, plays a default rollout policy from the new node, and propagates the simulated outcome to update the hyper-parameters for visited states and actions. Noting that the rollout policy is only played once for each new node at each iteration, the set of past observations $Z$ in the algorithm has size $n = 1$.

The function OnlinePlanning is the overall procedure interacting with the real environment. It is called with current state $s$, search tree $T$ initially empty and the maximal horizon $H$. It repeatedly calls the function DNG-MCTS until some resource budgets are reached (e.g., the computation is timeout or the maximal number of iterations is reached), by when a greedy action to be performed in the environment is returned to the agent.

### 3.5   The Convergency Property

For Thompson sampling in stationary MABs (i.e., the underlying reward function will not change), it is proved that: (1) the probability of selecting any suboptimal action $a$ at the current step is bounded by a linear function of the probability of selecting the optimal action; (2) the coefficient in this linear function decreases exponentially fast with the increase in the number of selection of optimal action [15]. Thus, the probability of selecting the optimal action in an MAB is guaranteed to converge to 1 in the limit using Thompson sampling.

**OnlinePlanning**$(s : state, T : tree, H : max\ horizon)$
Initialize $(\mu_{s,0}, \lambda_s, \alpha_s, \beta_s)$ for each $s \in S$
Initialize $\boldsymbol{\rho}_{s,a}$ for each $s \in S$ and $a \in A$
**repeat**
  |   **DNG-MCTS**$(s, T, H)$
**until** *resource budgets reached*
**return ThompsonSampling**$(s, H, False)$

**DNG-MCTS**$(s : state, T : tree, h : horizon)$
**if** $h = 0$ **or** *s is terminal* **then**
  ∟ **return** $0$
**else if** *node $\langle s, h \rangle$ is not in tree $T$* **then**
  |   Add node $\langle s, h \rangle$ to $T$
  |   Play rollout policy by simulation for $h$ steps
  |   Observe the outcome $r$
  ∟ **return** $r$
**else**
  |   $a \leftarrow$ **ThompsonSampling**$(s, h, True)$
  |   Execute $a$ by simulation
  |   Observe next state $s'$ and reward $R(s, a)$
  |   $r \leftarrow R(s, a) + \gamma$**DNG-MCTS**$(s', T, h - 1)$
  |   $\alpha_s \leftarrow \alpha_s + 0.5$
  |   $\beta_s \leftarrow \beta_s + (\lambda_s(r - \mu_{s,0})^2/(\lambda_s + 1))/2$
  |   $\mu_{s,0} \leftarrow (\lambda_s\mu_{s,0} + r)/(\lambda_s + 1)$
  |   $\lambda_s \leftarrow \lambda_s + 1$
  |   $\rho_{s,a,s'} \leftarrow \rho_{s,a,s'} + 1$
  ∟ **return** $r$

**ThompsonSampling**$(s : state, h : horizon, sampling : boolean)$
**foreach** $a \in A$ **do**
  ∟ $q_a \leftarrow$ **QValue**$(s, a, h, sampling)$
**return** $\arg\max_a q_a$

**QValue**$(s : state, a : action, h : horizon, sampling : boolean)$
$r \leftarrow 0$
**foreach** $s' \in S$ **do**
  |   **if** $sampling = True$ **then**
  |    ∟ Sample $w_{s'}$ according to $Dir(\boldsymbol{\rho}_{s,a})$
  |   **else**
  |    ∟ $w_{s'} \leftarrow \rho_{s,a,s'}/\sum_{n \in S} \rho_{s,a,n}$
  |   $r \leftarrow r + w_{s'}$**Value**$(s', h - 1, sampling)$
**return** $R(s, a) + \gamma r$

**Value**$(s : state, h : horizon, sampling : boolean)$
**if** $h = 0$ **or** *s is terminal* **then**
  ∟ **return** $0$
**else**
  |   **if** $sampling = True$ **then**
  |    |   Sample $(\mu, \tau)$ according to
  |    |   $NormalGamma(\mu_{s,0}, \lambda_s, \alpha_s, \beta_s)$
  |    ∟ **return** $\mu$
  |   **else**
  |    ∟ **return** $\mu_{s,0}$

Figure 1: Dirichlet-NormalGamma based Monte-Carlo Tree Search

The distribution of $X_{s,\pi}$ is determined by the transition function and the $Q$ values given the policy $\pi$. When the $Q$ values converge, the distribution of $X_{s,\pi}$ becomes stationary with the optimal policy. For the leaf nodes (level $H$) of the search tree, Thompson sampling will converge to the optimal actions with probability 1 in the limit since the MABs are stationary. When all the leaf nodes converge, the distributions of return values from them will not change. So the MABs of the nodes in level $H - 1$ become stationary as well. Thus, Thompson sampling will also converge to the optimal actions for nodes in level $H - 1$. Recursively, this holds for all the upper-level nodes. Therefore, we conclude that DNG-MCTS can find the optimal policy for the root node if unbounded computational resources are given.

## 4 Experiments

We have tested our DNG-MCTS algorithm and compared the results with UCT in three common MDP benchmark domains, namely *Canadian traveler problem*, *racetrack* and *sailing*. These problems are modeled as cost-based MDPs. That is, a cost function $c(s, a)$ is used instead of the reward function $R(s, a)$, and the min operator is used in the Bellman equation instead of the max operator. Similarly, the objective of solving a cost-based MDPs is to find an optimal policy that minimizes the expected accumulated cost for each state. Notice that algorithms developed for reward-based MDPs can be straightforwardly transformed and applied to cost-based MDPs by simply using the min operator instead of max in the Bellman update routines. Accordingly, the min operator is used in the function ThompsonSampling of our transformed DNG-MCTS algorithm. We implemented our codes and conducted the experiments on the basis of *MDP-engine*, which is an open source software package with a collection of problem instances and base algorithms for MDPs.[1]

Table 1: CTP problems with 20 nodes. The second column indicates the belief size of the trans-formed MDP for each problem instance. UCTB and UCTO are the two domain-specific UCT implementations [18]. DNG-MCTS and UCT run for 10,000 iterations. Boldface fonts are best in whole table; gray cells show best among domain-independent implementations for each group. The data of UCTB, UCTO and UCT are taken form [16].

| prob. | belief | domain-specific UCT | | random rollout policy | | optimistic rollout policy | |
|---|---|---|---|---|---|---|---|
| | | UCTB | UCTO | UCT | DNG | UCT | DNG |
| 20-1 | $20 \times 3^{49}$ | 210.7±7 | **169.0±6** | 216.4±3 | 223.9±4 | 180.7±3 | 177.1±3 |
| 20-2 | $20 \times 3^{49}$ | 176.4±4 | **148.9±3** | 178.5±2 | 178.1±2 | 160.8±2 | 155.2±2 |
| 20-3 | $20 \times 3^{51}$ | 150.7±7 | **132.5±6** | 169.7±4 | 159.5±4 | 144.3±3 | 140.1±3 |
| 20-4 | $20 \times 3^{49}$ | 264.8±9 | **235.2±7** | 264.1±4 | 266.8±4 | 238.3±3 | 242.7±4 |
| 20-5 | $20 \times 3^{52}$ | 123.2±7 | **111.3±5** | 139.8±4 | 133.4±4 | 123.9±3 | 122.1±3 |
| 20-6 | $20 \times 3^{49}$ | 165.4±6 | **133.1±3** | 178.0±3 | 169.8±3 | 167.8±2 | 141.9±2 |
| 20-7 | $20 \times 3^{50}$ | 191.6±6 | **148.2±4** | 211.8±3 | 214.9±4 | 174.1±2 | 166.1±3 |
| 20-8 | $20 \times 3^{51}$ | 160.1±7 | **134.5±5** | 218.5±4 | 202.3±4 | 152.3±3 | 151.4±3 |
| 20-9 | $20 \times 3^{50}$ | 235.2±6 | **173.9±4** | 251.9±3 | 246.0±3 | 185.2±2 | 180.4±2 |
| 20-10 | $20 \times 3^{49}$ | 180.8±7 | **167.0±5** | 185.7±3 | 188.9±4 | 178.5±3 | 170.5±3 |
| total | | 1858.9 | **1553.6** | 2014.4 | 1983.68 | 1705.9 | 1647.4 |

In each benchmark problem, we (1) ran the transformed algorithms for a number of iterations from the current state, (2) applied the best action based on the resulted action-values, (3) repeated the loop until terminating conditions (e.g., a goal state is satisfied or the maximal number of running steps is reached), and (4) reported the total discounted cost. The performance of algorithms is evaluated by the average value of total discounted costs over 1,000 independent runs. In all experiments, $(\mu_{s,0}, \lambda_s, \alpha_s, \beta_s)$ is initialized to $(0, 0.01, 1, 100)$, and $\rho_{s,a,s'}$ is initialized to 0.01 for all $s \in S$, $a \in A$ and $s' \in S$. For fair comparison, we also use the same settings as in [16]: for each decision node, (1) only applicable actions are selected, (2) applicable actions are forced to be selected once before any of them are selected twice or more, and 3) the exploration constant for the UCT algorithm is set to be the current mean action-values $Q(s, a, d)$.

The *Canadian traveler problem* (CTP) is a path finding problem with imperfect information over a graph whose edges may be blocked with given prior probabilities [17]. A CTP can be modeled as a deterministic POMDP, i.e., the only source of uncertainty is the initial belief. When transformed to an MDP, the size of the belief space is $n \times 3^m$, where $n$ is the number of nodes and $m$ is the number of edges. This problem has a discount factor $\gamma = 1$. The aim is to navigate to the goal state as quickly as possible. It has recently been addressed by an anytime variation of AO*, named AOT [16], and two domain-specific implementations of UCT which take advantage of the specific MDP structure of the CTP and use a more informed base policy, named UCTB and UCTO [18]. In this experiment, we used the same 10 problem instances with 20 nodes as done in their papers.

When running DNG-MCTS and UCT in those CTP instances, the number of iterations for each decision-making was set to be 10,000, which is identical to [16]. Two types of default rollout policy were tested: the *random policy* that selects actions with equal probabilities and the *optimistic policy* that assumes traversability for unknown edges and selects actions according to estimated cost. The results are shown in Table 1. Similar to [16], we included the results of UCTB and UCTO as a reference. From the table, we can see that DNG-MCTS outperformed the domain-independent version of UCT with random rollout policy in several instances, and particularly performed much better than UCT with optimistic rollout policy. Although DNG-MCTS is not as good as domain-specific UCTO, it is competitive comparing to the general UCT algorithm in this domain.

The *racetrack* problem simulates a car race [19], where a car starts in a set of initial states and moves towards the goal. At each time step, the car can choose to accelerate to one of the eight directions. When moving, the car has a possibility of 0.9 to succeed and 0.1 to fail on its acceleration. We tested DNG-MCTS and UCT with random rollout policy and horizon $H = 100$ in the instance of *barto-big*, which has a state space with size $|S| = 22534$. The discount factor is $\gamma = 0.95$ and the optimal cost produced is known to be 21.38. We reported the curve of the average cost as a function of the number of iterations in Figure 2a. Each data point in the figure was averaged over 1,000

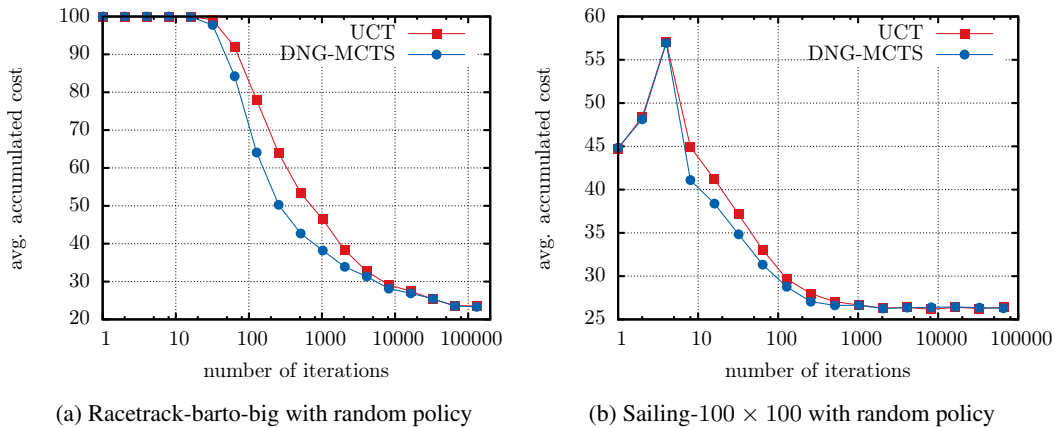

<center>(a) Racetrack-barto-big with random policy      (b) Sailing-$100 \times 100$ with random policy</center>

<center>Figure 2: Performance curves for Racetrack and Sailing</center>

runs, each of which was allowed for running at most 100 steps. It can be seen from the figure that DNG-MCTS converged faster than UCT in terms of sample complexity in this domain.

The *sailing* domain is adopted from [3]. In this domain, a sailboat navigates to a destination on an 8-connected grid. The direction of the wind changes over time according to prior transition probabilities. The goal is to reach the destination as quickly as possible, by choosing at each grid location a neighbour location to move to. The discount factor in this domain is $\gamma = 0.95$ and the maximum horizon is set to be $H = 100$. We ran DNG-MCTS and UCT with random rollout policy in a $100 \times 100$ instance of this domain. This instance has 80000 states and the optimal cost is 26.08. The performance curve is shown in Figure 2b. A trend similar to the racetrack problem can be observed in the graph: DNG-MCTS converged faster than UCT in terms of sample complexity.

Regarding computational complexity, although the total computation time of our algorithm is linear with the total sample size, which is at most $width \times depth$ ($width$ is the number of iterations and $depth$ is the maximal horizon), our approach does require more computation than simple UCT methods. Specifically, we observed that most of the computation time of DNG-MCTS is due to the sampling from distributions in Thompson sampling. Thus, DNG-MCTS usually consumes more time than UCT in a single iteration. Based on our experimental results on the benchmark problems, DNG-MCTS typically needs about 2 to 4 times (depending on problems and the iterating stage of the algorithms) of computational time more than UCT algorithm for a single iteration. However, if the simulations are expensive (e.g., computational physics in 3D environment where the cost of executing the simulation steps greatly exceeds the time needed by action-selection steps in MCTS), DNG-MCTS can obtain much better performance than UCT in terms of computational complexity because DNG-MCTS is expected to have lower sample complexity.

## 5 Conclusion

In this paper, we proposed our DNG-MCTS algorithm — a novel Bayesian modeling and inference based Thompson sampling approach using MCTS for MDP online planning. The basic assumption of DNG-MCTS is modeling the uncertainty of the accumulated reward for each state-action pair as a mixture of Normal distributions. We presented the overall Bayesian framework for representing, updating, decision-making and propagating of probability distributions over rewards in the MCTS search tree. Our experimental results confirmed that, comparing to the general UCT algorithm, DNG-MCTS produced competitive results in the *CTP* domain, and converged faster in the domains of *racetrack* and *sailing* with respect to sample complexity. In the future, we plan to extend our basic assumption to using more complex distributions and test our algorithm on real-world applications.

<center>8</center>

## Acknowledgements

This work is supported in part by the National Hi-Tech Project of China under grant 2008AA01Z150 and the Natural Science Foundation of China under grant 60745002 and 61175057. Feng Wu is supported in part by the ORCHID project (`http://www.orchid.ac.uk`). We are grateful to the anonymous reviewers for their constructive comments and suggestions.

## Footnotes

[1]MDP-engine can be publicly accessed via `https://code.google.com/p/mdp-engine/`

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
