[Reviews · NeurIPS 2013]

Submitted by Assigned_Reviewer_5

The paper proposes a Bayesian inference in Monte-Carlo tree search (MCTS) with Thompson sampling based action-selection strategy, called Dirichlet-NormalGamma MCTS (DNG-MTCS) algorithm. The method approximates the accumulated reward of following the current policy from a state, X_{s,\pi(s)}, by the normal distribution with the NromalGamma distribution prior. The state transition probabilities are estimated via Dirichlet distributions. Action-selection strategy is based on Thompson sampling approach, where the expected cumulative reward for each action is computed with the parametric distribution with parameters drawn from the posterior distributions and then the action with the highest expectation is selected. The authors apply the proposed method to several benchmark tasks and showed that the method can converge (slightly) faster than the UCT algorithm. Theoretical properties about convergence are also provided.

Although there is a lot of work on MCTS, I am not aware of other work based on the specific approach made in the paper. However, the density approximation of the cumulative reward with NormalGamma prior is not new and has been proposed by Dearden et al. 1998 [a]. Modeling of the state transition with Dirichlet prior is well known. So the originality of the proposed modeling and inference methods in Section 3.2 will be small. However, the proposed combination with this modeling and action-selection strategy based on Thompson sampling is new, to the best of my knowledge, and might be of interest to related areas of research.

The presented experimental result would be week. Although there is a lot of work on MCTS, the applied baseline method was limited to the classic UCT (except for the CTP problem). More detail experiments in terms of computational costs as well as sample complexity with several advanced methods in [b,d,e] will be needed.

Most part of this paper is clearly written. But the assumptions in Section 3.1 are not quite clear. I feel that there is a contradiction. Although the cumulative reward given (s, a), X_{s,a}, is modeled as a mixture of Normal distributions as a result, the authors argue the assumption of X_{s,\pi(s)} having a Normal distribution, is realistic. However, in what some see as, X_{s,\pi(s)} has to be a mixture of Normal distributions and the number of mixtures will increase exponentially (from leaf states to root). So their assumption of X_{s,\pi(s)} cannot be realistic. But I think it should be one of practical approaches for approximating the distribution of X_{s,\pi(s)}. The relation between cumulative rewards in successive time-steps would be related to the distributional Bellman equation in [c]. I’d like to recommend revising this part carefully in the final version.

In line 137--139, N(n\mu, \delta^2/n) is N(n\mu, n\delta^2)? Doesn't the \sum_n f diverge as n \to \infty?


[a] Richard Dearden, Nir Friedman, Stuart J. Russell: Bayesian Q-Learning. AAAI/IAAI 1998.
[b] Gerald Tesauro, V. T. Rajan, Richard Segal: Bayesian Inference in Monte-Carlo Tree Search. UAI 2010.
[c] Tetsuro Morimura, Masashi Sugiyama, Hisashi Kashima, Hirotaka Hachiya, Toshiyuki Tanaka: Parametric Return Density Estimation for Reinforcement Learning. UAI 2010.
[d] John Asmuth and Michael L. Littman. Approaching Bayes-optimalilty using Monte-Carlo tree search. In Proc. 21st Int. Conf. Automat. Plan. Sched., 2011.
[e] Arthur Guez, David Silver, Peter Dayan: Efficient Bayes-Adaptive Reinforcement Learning using Sample-Based Search. NIPS 2012.
Summary: The paper presents a Bayesian MCTS with Thompson-sampling-based action-selection strategy. It is shown through several benchmark tasks that the proposed method can converge faster than the UCT algorithm.

Submitted by Assigned_Reviewer_8

This paper provides a novel addition to the large and still-growing body of MCTS research. The idea is to represent the distribution over accumulated rewards in the search tree as a mixture of Gaussians by using a Dirichlet as a prior on mixture weights, and a NormalGamma as a prior for the mixtures themselves. The authors justify (using the central limit theorem) their assumptions of normality and give a brief overview of setting the hyperparameters of the priors. This Bayesian formulation is combined with Thompson sampling to provide a nice version of MCTS that seems to avoid the more heuristic nature of UCB and be more robust in its explore/exploit tradeoff.

The structure and presentation of the paper are good, and the ideas and math are presented clearly. There are language issues throughout, but not enough to make it unintelligible (although these should be cleaned up -- especially in the introduction). The paper does a good job of introducing the requisite concepts, stating and justifying the underlying modeling assumptions, presenting the model, and then putting it all together -- though a more descriptive comparison between Thompson sampling and UCB would be good, especially their specific assumptions and strengths/weaknesses.

The empirical evaluation is solid; however, the authors should include a more thorough comparison of computational complexity with UCT (with numbers / figures). Clearly, this algorithm is more complex than UCT and so will run slower, but how much slower? Seconds? Hours? Days? One of the reasons UCT is so effective is that it's extremely fast and enables much more sampling in the same amount of time as more complicated methods, thus providing more accurate empirical estimates, even if they're not computed as cleverly. Also, I'm curious how much benefit came from just Thompson sampling and how much from using the Dirichlet-NormalGamma distribution versus vanilla UCT. Is there some way to quantify these gains separately?

A question I have, regarding the assumptions made, is that these claims of convergence to a Normal rely on the fact that the future policy is fixed and thus the rollout is a Markov chain. However, as I understand it, this policy is not fixed and changes over time (either through learning or exploratory randomness). Doesn't this invalidate the claims made? (though, empirically, it seems like the assumptions aren't that harmful)
Summary: This is a solid paper that presents a small novel improvement to Monte carlo tree search in the form of using Bayesian mixture modeling to represent the accumulated reward of actions in an MDP. The idea is solid, explained well, and empirically validated but the paper has minor issues in terms of language, computational complexity, and, possibly, the validity of certain assumptions.

Submitted by Assigned_Reviewer_10

The authors present a Monte-Carlo tree search algorithm for online planning in MDPs with unknown transition probabilities. They use Thompson sampling, which is known to be a randomized Bayesian algorithm to minimize regret in the multi-armed stochastic bandit problems, for the action selection to expand the search tree. They also present the probability of the return X_{s,\pi(s)} and X_{s,a} as a normal distribution and a mixture of normal distributions respectively in order to use Thompson sampling, which chooses an action according to the posterior probability of being the best action.

I think it is a good idea to exploit Thompson sampling in Monte-Carlo tree search since some recent studies have empirically showed that Thompson sampling outperforms a popular alternative, UCB. The problem of using Thompson sampling in MDPs is modeling the probability of the returns in the Bayesian setting, which is addressed in this paper. However I have a few questions.

(1) I think it is ok to present the distribution of the return X_{s,\pi(s)} as a normal distribution when X_{s,\pi(s)} denotes the leaf node in the search tree and \pi is the roll-out policy. However, the other nodes in the upper levels do not satisfy the authors’ claim since the policy changes.
(2) The authors say that “Thompson sampling converges to find the optimal action with probability 1 by treating each decision node in the tree as MAB” in lines 263-264, but I cannot believe this statement. The MAB problems in the decision nodes except for the leaf nodes seem to be non-stationary because the policies change as the returns X_{s,\pi(s)} change. As far as I know, there is no convergence guarantee of Thompson sampling in non-stationary MAB problems. Therefore, I am not sure that the policy found by DNG-MCTS converges to the optimal one.
(3) Does the word “iteration” in the experiments section (lines 315-316, 354-355, 368) denote the iteration of DNG-MCTS in the function “OnlinePlanning” in figure 1?
(4) In the last paragraph of the experiments section, the authors say that DNG-MCTS requires more computation time than UCT. It would be good to report the CPU running time.

Minor comments: There are some errors in the references. Some papers such as [7, 8, 15, 18] have been already published in some conferences rather than arXiv.

Quality: I think this paper has a few flaws.

Clarity: The paper is well written as easy to follow.

Originality: The paper is original to my knowledge.

Significance: The main idea of the paper is quite interest to the RL community but I think the authors should clarify a few flaws.

-----
The original central limit theorem (CLT) for Markov chains is \sqrt{n}(n^(-1) \sum_{t=0}^n f(s_t) - E[f(s_t)]) \rightarrow N(0,\sigma^2) as n \rightarrow \infty. It says that the empirical mean of f(s_t) converges to the true expectation of f(s_t). In Section 3.1, the authors slightly modified the CLT and showed that \frac{1}{\sqrt{n}}(\sum_{t=0}^n f(s_t) - n\mu) \rightarrow N(0,\sigma^2). As the reviewer#5 already pointed out, I think that the sum of f(s_t) follows N(n\mu, n\sigma^2) and the sum of f(s_t) diverges as n goes to infinity. Thus it is difficult for me to accept that the CLT justifies the Normal assumption on the sum of rewards although the assumption seems to be practically reasonable.
Summary: The authors provide a novel MCTS algorithm by using Thompson sampling, which is quite interest to the RL community, but the soundness is somewhat lacking.
Author Feedback

Author rebuttal: We thank all the reviewers for their valuable comments and suggestions.

Regarding the convergence, please note that the distribution of X_{s, \pi(s)} is determined by the transition function and the Q value given the policy \pi. When the Q value converges, the distribution of X_{s, \pi(s)} becomes stationary with the optimal policy. For the leaf nodes (level H) of the search tree, Thompson sampling will converge to the optimal actions with probability 1 in the limit since the MABs are stationary (as pointed out by Reviewer 1). When all the leaf nodes converge, the distributions of return values from them will *not* change. So the MABs of the nodes in level H-1 become stationary as well. Thus, Thompson sampling will also converge to the optimal actions for nodes in level H-1. Recursively, this holds for all the upper-level nodes. Therefore, we conclude that DNG-MCTS can find the optimal policy for all the nodes if unbounded computational resources are given.

Regarding the assumption, X_{s, \pi(s)} as a Normal distribution is a convenient and reasonable approximation of its real distribution. As explained in Section 3.1, for a given policy \pi, an MDP reduces to a stationary Markov chain. Thus, according to the central limit theorem for Markov chains (Equation 1), we can approximate X_{s, \pi(s)} as a Normal distribution (if H is sufficiently large). Before the policy \pi converges, the real distribution of X_{s, \pi(s)} is unknown and could be a mixture of Normal distributions (as pointed out by Reviewer 2). However, when the policy \pi converges to the optimal solution, it is reasonable to approximate X_{s, \pi(s)} as a Normal distribution because the return values from state s given the optimal policy is approximately normally distributed (according to Equation 1) with the expectation be the optimal value function of state s. As shown in the experimental results, it works quite well in practice. We will revise this part carefully as suggested.

Regarding the computational time, based on our experimental results on the benchmark problems, DNG-MCTS typically needs about 2 to 4 times (depending on problems and the iterating stage of the algorithms) of computational time more than UCT algorithm for a single iteration. As pointed out in the last paragraph of Section 4, if the simulations are expensive, DNG-MCTS can obtain much better performance than UCT in terms of computational complexity because DNG-MCTS is expected to have lower sample complexity.

Regarding Q3 of Reviewer 1, the word “iteration” in the experiments section denotes the iteration of DNG-MCTS in the function “OnlinePlanning” in Figure 1. All the typos and the errors in the references will be corrected as suggested by Reviewer 1.